# Prognostic Effect of a Novel Simply Calculated Nutritional Index in Acute Decompensated Heart Failure

**DOI:** 10.3390/nu12113311

**Published:** 2020-10-29

**Authors:** Sayaki Ishiwata, Shoichiro Yatsu, Takatoshi Kasai, Akihiro Sato, Hiroki Matsumoto, Jun Shitara, Megumi Shimizu, Azusa Murata, Takao Kato, Shoko Suda, Shinichiro Doi, Masaru Hiki, Yuya Matsue, Ryo Naito, Hiroshi Iwata, Atsutoshi Takagi, Hiroyuki Daida

**Affiliations:** 1Department of Cardiovascular Medicine, Juntendo University School of Medicine, Tokyo 113-8421, Japan; s-ishiwata@juntendo.ac.jp (S.I.); syatsu@juntendo.ac.jp (S.Y.); ak-sato@juntendo.ac.jp (A.S.); hmatsumo@juntendo.ac.jp (H.M.); jshitara@juntendo.ac.jp (J.S.); megumi-s@juntendo.ac.jp (M.S.); azmurata@juntendo.ac.jp (A.M.); tkatou@juntendo.ac.jp (T.K.); ssuda@juntendo.ac.jp (S.S.); doies@juntendo.ac.jp (S.D.); ma-hiki@juntendo.ac.jp (M.H.); yuya8950@gmail.com (Y.M.); rnaitou@juntendo.ac.jp (R.N.); h-iwata@juntendo.ac.jp (H.I.); a.taka@juntendo.ac.jp (A.T.); daida@juntendo.ac.jp (H.D.); 2Cardiovascular Respiratory Sleep Medicine, Juntendo University Graduate School of Medicine, Tokyo 113-8421, Japan; 3Sleep and Sleep-Disordered Breathing Center, Juntendo University Hospital, Tokyo 113-8421, Japan

**Keywords:** acute decompensated heart failure, nutritional index, prognosis

## Abstract

The TCB index (triglycerides × total cholesterol × body weight), a novel simply calculated nutritional index based on serum triglycerides (TGs), serum total cholesterol (TC), and body weight (BW), was recently reported to be a useful prognostic indicator in patients with coronary artery disease. Thus, this study aimed to investigate the relationship between TCBI and long-term mortality in acute decompensated heart failure (ADHF) patients. Patients with a diagnosis of ADHF who were consecutively admitted to the cardiac intensive care unit in our institution from 2007 to 2011 were targeted. TCBI was calculated using the formula TG (mg/dL) × TC (mg/dL) × BW (kg)/1000. Patients were divided into two groups according to the median TCBI value. An association between admission TCBI and mortality was assessed using univariable and multivariable Cox proportional hazard analyses. Overall, 417 eligible patients were enrolled, and 94 (22.5%) patients died during a median follow-up period of 2.2 years. The cumulative survival rate with respect to all-cause, cardiovascular, and cancer-related mortalities was worse in patients with low TCBI than in those with high TCBI. In the multivariable analysis, although TCBI was not associated with cardiovascular and cancer mortalities, the association between TCBI and reduced all-cause mortality (hazard ratio: 0.64, 95% confidence interval: 0.44–0.94, *p* = 0.024) was observed. We computed net reclassification improvement (NRI) when TCBI or Geriatric Nutritional Risk Index (GNRI) was added on established predictors such as hemoglobin, serum sodium level, and both. TCBI improved discrimination for all-cause mortality (NRI: 0.42, *p* < 0.001; when added on hemoglobin and serum sodium level). GNRI can improve discrimination for cancer mortality (NRI: 0.96, *p* = 0.002; when added on hemoglobin and serum sodium level). TCBI, a novel and simply calculated nutritional index, can be useful to stratify patients with ADHF who were at risk for worse long-term overall mortality.

## 1. Introduction

Heart failure (HF) is one of the major causes of mortality, although the treatment of HF has been improving. As the aging population of Japan increases, the prevalence of malnutrition in patients with HF has become more prominent. Malnutrition is associated with an increase in mortality rate in patients with chronic cardiovascular diseases such as peripheral artery disease, coronary artery disease, and chronic HF [1]. Some objective nutritional indicators such as the Geriatric Nutritional Risk Index (GNRI), the Controlling Nutritional Status (CONUT), and the prognostic nutritional index (PNI) are well known, and the relationship between these indicators and cardiovascular diseases has been investigated [2,3,4]. In particular, in patients with chronic HF, these nutritional indices were assessed, and malnutrition was reported to contribute to worsening long-term clinical outcomes [5]. In acute decompensated HF (ADHF) patients, acute congestion could dilute body fluids and influence some laboratory parameters; thus, simple indicators such as serum albumin and cholesterol might not be useful in predicting long-term prognosis. Although the prognostic effect of multiple indicators has also been studied in patients with ADHF [6], these indicators have not frequently been used in cardiovascular clinical practice because of their complex calculation methods.

The novel and simply calculated nutritional index, the TCB index (TCBI), calculated by multiplying serum triglycerides (TGs, mg/dL), serum total cholesterol (TC, mg/dL), and body weight (BW, kg) has been implicated as a useful prognostic indicator in patients with coronary artery disease and critically ill patients [7,8]. However, the association of TCBI with long-term outcomes in patients with ADHF has not been elucidated.

To illustrate the usefulness of TCBI, a comparison between other indicators proven valuable for predicting long-term outcomes and TCBI is warranted. In general, malnutrition assessed by GNRI has been reported to be associated with increased cancer mortality in patients with malignancy [9,10], and our group previously reported that the nutritional status based on GNRI was useful for stratifying ADHF patients who was at risk for prolonged hospital stay in association with HF with preserved ejection fraction [11]. However, the importance of TCBI in patients with ADHF remains to be elucidated. This study aimed to investigate whether TCBI is useful as a prognostic predictor in patients with ADHF.

## 2. Materials and Methods

### 2.1. Subjects

This is a retrospective observational cohort analysis, and patients who were hospitalized in the cardiac intensive care unit at Juntendo University Hospital, Tokyo, Japan between 2007 and 2011 with a diagnosis of ADHF were enrolled. ADHF was defined based on the modified Framingham criteria [12,13], only including variables estimated at admission in the Framingham criteria [14]. Patients who were <18 years old, had an acute coronary syndrome and/or had undergone cardiac surgery during the previous 4 weeks or during initial hospitalization, had end-stage renal disease requiring dialysis, and diagnosed with a life-threatening malignancy were excluded from the study. Patients whose TG, TC, and BW were not measured on admission were also excluded. The Institutional Review Board of the Juntendo University Hospital approved the study protocol (871), and the study complied with the Declaration of Helsinki. Informed consent was obtained from all patients.

### 2.2. Data Collection

Baseline data were collected prospectively at initial hospitalization. Baseline biochemical parameters were obtained in the first 24 h after admission during the fasting state. A clinical chart review of all patients was conducted to obtain medical history. Diabetes mellitus was defined as an HbA1c level greater than 6.5 or patients who were prescribed antidiabetic agents such as oral hypoglycemic agents or insulin. Renal function was shown as estimated glomerular filtration rate (eGFR) which was calculated by the Modification of Diet in Renal Disease equation with a Japanese coefficient from baseline serum creatinine levels [15]. Echocardiography was performed for every patient within 24 h after admission, including two-dimensional echocardiographic images and Doppler flow. The left ventricular ejection fraction was obtained using the modified Simpson’s method in each patient. TG was measured by enzyme colorimetric method. TC was measured by the enzyme method. Albumin was measured by the improved BCP method. TCBI was calculated using TG, TC, and BW using the following formula: TCBI = TG (mg/dL) × TC (mg/dL) × BW (kg)/1000 [7]. All eligible patients were divided into two groups in accordance with the median TCBI. As a control nutritional index for comparison with TCBI, GNRI was calculated in accordance with the following formula as previously reported [16]: GNRI = 14.9 × serum Albumin (g/dL) + 41.7 × body mass index (BMI)/22. We followed all patients from the date of index admission until July 2013. Outcome data were obtained by reviewing the hospital medical records for all deaths recorded following discharge. The endpoints of interest were three types of mortality in the follow-up period: all-cause mortality, cardiovascular mortality, and cancer mortality.

### 2.3. Statistical Analysis

Continuous variables are expressed as the mean ± standard deviation or median with interquartile range. Categorical variables are presented as numbers and percentages. Comparing the baseline characteristics between the two groups, the χ^2^ test or Fisher’s exact test was used for categorical variables, while a *t*-test or Mann–Whitney U test was used for continuous variables. To assess a correlation between TCBI and GNRI, the Spearman rank correlation coefficient was computed. The event-free survival curves were presented using the Kaplan–Meier method and compared between groups with < or ≥ median value of TCBI by log-rank test to evaluate the relationship between TCBI and the three types of mortality. The univariable Cox proportional hazards regression analysis was used to identify the association between the three types of mortality and the variables on admission, including age, gender (male), medical history (i.e., ischemic heart diseases, history of HF, diabetes mellitus, and atrial fibrillation), mean blood pressure (MBP), left ventricular ejection fraction (LVEF), laboratory tests on admission (i.e., brain natriuretic peptide [BNP], blood urea nitrogen (BUN), hemoglobin, eGFR, sodium, C-reactive protein (CRP)), medications during hospitalization (i.e., diuretics, angiotensin-converting-enzyme inhibitor [ACE-I]/angiotensin receptor blocker (ARB), β-blocker, and spironolactone)), GNRI and TCBI. The natural log-transformed values were used for TCBI, GNRI, CRP, and plasma BNP, as these values were skewed. These variables with *p*-values less than 0.1 in each univariable analysis were included in the multivariable analysis. Multivariable analysis was established to identify the independent predictors for all-cause, cardiovascular, and cancer mortality based on backward stepwise Cox proportional hazards regression analysis. Receiver operating characteristic (ROC) curves were drawn, and areas under the curve (AUCs) were measured for each mortality; we compared the two AUCs. Furthermore, the cutoff value of TCBI with positive and negative predictive values was determined. We computed the net reclassification improvement (NRI) when TCBI or GNRI was added on the established predictors such as hemoglobin and serum sodium level. A *p*-value of 0.05 was considered statistically significant. All analyses were performed using a statistical software package (JMP ver. 11.0, SAS Corporation, Cary, NC, USA).

## 3. Results

### 3.1. Baseline Characteristics of Patients with TCBI ≥ 745 and TCBI < 745

Overall, 751 patients with ADHF were admitted to our institution between 2007 and 2011. Among them, 190 patients with concomitant acute coronary syndrome and/or who had undergone cardiac surgery during the previous 4 weeks, end-stage renal disease requiring dialysis, and a life-threatening malignancy were initially excluded. In addition, 144 patients whose TG, TC, and BW were not measured on admission were also excluded. Ultimately, a total of 417 patients were enrolled and classified into two groups according to the median TCBI score (i.e., 745). Of these, 208 patients (49.9%) were classified into the lower TCBI group (TCBI < 745). Baseline characteristics were shown in Table 1. Compared to those with TCBI ≥ 745 (higher TCBI group), the lower TCBI group was older (74.2 ± 11.0 and 65.7 ± 14.2, *p* < 0.001) and had a lower BMI (21.2 ± 3.7 and 24.6 ± 5.4, *p* < 0.001), BW (52.7 ± 12.4 and 65.8 ± 18.5, *p* < 0.001), TC (147 ± 31.3 and 191 ± 41.7, *p* < 0.001), TG (64.2 ± 19.8 and 132.2 ± 71.6, *p* < 0.001), hemoglobin (11.8 ± 2.2 and 13.2 ± 2.4, *p* < 0.001), and sodium levels (138.2 ± 4.4 and 139.2 ± 3.8, *p* < 0.001). In the lower TCBI group, the prevalence of atrial fibrillation was higher (94 [44.9%] and 91 [34.1%]), *p* = 0.027) while diabetes mellitus was lower (61 [29.1%] and 90 [43.2%], *p* = 0.003) than those in the higher TCBI group. In terms of medications during hospitalization, the use of diuretics (41.6% and 29.8%, *p* = 0.014) and aldosterone antagonists (16.7% and 9.1%, *p* = 0.028) was more common in the lower TCBI group than in the higher TCBI group.

### 3.2. Nutritional Indexes and Mortality

The median follow-up period was 2.4 years. During follow-up, the all-cause mortality was observed in 94 out of 417 patients (23%). Among the 94 patients, 57 (61%) died from cardiovascular causes, while 7 (7%) were cancer-related deaths. The event-free survival curves showed that the cumulative incidence of the all-cause, cardiovascular, and cancer mortality was significantly higher in patients with lower TCBI (log-rank test, all-cause mortality: *p* < 0.001; cardiovascular mortality: *p* = 0.041; and cancer mortality: *p* = 0.031) (Figure 1a–c).

TCBI and GNRI were directly correlated; however, this correlation was rendered weak (correlation coefficient 0.29, *p* < 0.001) (Figure 2). In univariable Cox proportional hazard analysis, age, gender (male), ischemic heart disease, atrial fibrillation, MBP, hemoglobin, eGFR, sodium, blood urea nitrogen, log CRP, log BNP, history of HF, diuretics, log GNRI and log TCBI were associated with the all-cause mortality (Table 2a). In terms of cardiovascular mortality, age, hemoglobin, sodium, BUN, diuretics, aldosterone antagonists, beta-blockers, log BNP, log GNRI and log TCBI were significant predictors in the univariable analysis (Table 2b). Conversely, hemoglobin, LVEF, GNRI and TCBI were associated with cancer mortality in the univariable analysis (Table 2c).

To compare the discriminatory power of TCBI with that of GNRI in terms of long-term mortality risk, we computed the area under the ROC curves of TCBI and GNRI in each outcome. In all-cause mortality, the AUC of TCBI and GNRI were 0.640 (*p* < 0.001) and 0.646 (*p* < 0.001), respectively. In cardiovascular mortality, the AUC of TCBI and GNRI were 0.602 (*p* = 0.012) and 0.584 (*p* = 0.052), respectively. In terms of cancer mortality, the AUC of TCBI and GNRI were 0.745 (*p* = 0.038) and 0.751 (*p* = 0.019), respectively. Both TCBI and GNRI were promoted as useful indicators of mortality risk and Delong’s test revealed no statistically significant differences between TCBI and GNRI in each mortality (all-cause death: *p* = 0.989, cardiovascular death: *p* = 0.635, cancer death: *p* = 0.983) (Figure 3).

We computed the areas under the ROC curves of hemoglobin and serum sodium levels as well. In all-cause mortality, the AUC of hemoglobin was greater than that of the TCBI (0.743 vs. 0.626, *p* < 0.001). In cardiovascular mortality, the AUC of hemoglobin for cardiovascular death was greater than that of the TCBI (0.720 vs. 0.581, *p* = 0.004). However, there is no significant difference in AUC for cancer death between hemoglobin and TCBI (0.718 vs. 0.748, *p* = 0.579). In terms of serum sodium level, there are no significant differences in any AUC for all-cause, cardiovascular and cancer deaths between serum sodium level and TCBI (all-cause death: 0.604 for sodium vs. 0.624 for TCBI, *p* = 0.650; cardiovascular death: 0.570 for sodium vs. 0.572 for TCBI, *p* = 0.971; cancer death: 0.670 for sodium vs. 0.821 for TCBI, *p* = 0.212).

In the multivariable analysis, variables associated with the all-cause mortality included TCBI (HR 0.64, 98% CI 0.44–0.94, *p* = 0.024) along with age, hemoglobin, serum sodium level, BUN, and diuretics (Table 2a). Moreover, we found that TCBI of 578 as the cutoff value to detect all-cause mortality with positive/negative predictive values of 0.844 and 0.341, respectively. In terms of cardiovascular mortality, TCBI was not found to be a significant variable in the multivariable analysis (Table 2b). In addition to the LVEF, GNRI was a significant predictor of cancer mortality (HR 0.01, 98% CI 0.01–0.1, *p* = 0.006) (Table 2c).

We calculated NRI when the TCBI or GNRI was added onto hemoglobin, serum sodium level and both. In all-cause mortality, we found that adding log TCBI on hemoglobin resulted in significant NRI (0.28, *p* = 0.016), and that log TCBI onto serum sodium level resulted in significant NRI (0.34, *p* < 0.001). Furthermore, adding log TCBI onto hemoglobin and serum sodium level also resulted in significant NRI (0.42, *p* < 0.001). In cancer mortality, we found that adding log GNRI onto hemoglobin resulted in significant NRI (0.69, *p* = 0.046), and that adding log GNRI onto serum sodium level also resulted in significant NRI (0.94, *p* < 0.001). Furthermore, adding log GNRI onto hemoglobin and serum sodium level resulted in significant NRI (0.96, *p* = 0.002).

## 4. Discussion

The findings of the present study in which a single-center observational cohort of 417 patients hospitalized due to ADHF that were retrospectively analyzed provide several novel insights into the relationship between nutritional assessment and clinical outcomes in patients with HF. First, we found that in patients with ADHF, the correlation between TCBI, which is a novel simply calculated nutritional index, and GNRI, which is a commonly used nutritional index, was only weak, indicating that TCBI may have information for nutritional assessment distinct from GNRI. Second, a lower TCBI score was associated with an increased risk of all-cause death and a lower GNRI score was associated with that of cancer death in the multivariable analyses, while no association was found between either TCBI or GNRI and the cardiovascular mortality risk. Third, the AUC of the receiver operating characteristic curves showed similar discriminatory power of TCBI to GNRI in terms of mortality risk. Finally, in analyses regarding NRI when adding the TCBI or GNRI onto established predictors such as hemoglobin, serum sodium level and both, TCBI improved the discrimination of all-cause mortality risk and GNRI improved the discrimination of cancer mortality risk.

Being overweight (BMI > 25–30 kg/m^2^) and obese (BMI > 30 kg/m^2^) have been reported as long-term risk factors for the development of HF [17]. However, overweight and obesity contribute to better long-term prognosis in elderly patients with HF, known as the obesity paradox [1]. TC is known as a risk factor for atherosclerosis and also a nutritional indicator of chronic diseases. The reduction in TC has been shown to be associated with poor prognosis in chronic HF patients [18]. Furthermore, lower TC levels were shown as an independent predictor for increasing in-hospital mortality in patients with ADHF [19]. TG have also been observed as a possible objective nutritional index [20]. One study examining a total 637 patients with chronic HF revealed that TG were independently correlated with cardiovascular death only in women [21]. However, the long-term prognostic effect of TC and TG on ADHF is not well established. In addition, lipid-lowering medication was used in most patients with cardiovascular disease, and we should consider the effect of these medications. Serum albumin was reported as one of the simple nutritional indicators and hypoalbuminemia was associated with a poor prognosis in chronic HF patients [22,23]. In the study of patients admitted with ADHF, conflicting results were described in terms of the association between hypoalbuminemia and long-term outcomes [24,25]. However, using albumin as a prognostic indicator of HF may pose as a limitation because data concerning single indicators in the acute phase of HF is not associated with a long-term prognosis because the value may fluctuate during hospitalization [26].

To indemnify the disadvantage posed by a single nutritional indicator, complex indicators were introduced in patients with HF. GNRI score is calculated using the serum albumin level and BW, and malnutrition defined by GNRI was a useful prognostic indicator in chronic HF patients [5]. GNRI needs a more complex calculation than TCBI because it incorporates the BMI. In 490 acute HF patients older than 65 years, GNRI was well validated and associated with all-cause deaths [27]. CONUT score is obtained using the serum albumin level, TC level, and total lymphocyte count, and in a study using the CONUT score for assessing long-term prognosis in a total of 482 chronic HF patients, it was reported as a useful prognostic indicator for all-cause deaths [28]. Another study involving 635 patients with acute HF revealed that the increase in CONUT score was associated with all-cause deaths in a multivariable analysis [29]. PNI is composed of serum albumin and lymphocyte count. One previous study investigated the prognostic effect of using PNI on 388 chronic HF patients and demonstrated that PNI had a significant association with cardiovascular death and readmission in chronic HF patients [5]. In addition, one study that investigated 1673 acute HF patients with either reduced or preserved EF showed that PNI was associated with both all-cause and cardiovascular deaths, despite the substantially decreased levels of lymphocyte count and serum albumin [30]. Meanwhile, one recent report showed that nutritional indices, such as CONUT, PNI, GNRI, and subjective global assessment, were significantly associated with a 1-year mortality in patients with ADHF, and PNI might have the strongest predictive power among these [2]. However, these nutritional indices entail complex calculations.

In previous studies, TCBI was shown as a useful predictor of long-term outcomes in some cardiovascular disease populations. In 3567 patients who underwent percutaneous coronary intervention, TCBI was associated with a reduced all-cause, cardiovascular, and cancer mortality [7]. In hospitalized patients with critical cardiovascular disease using mechanical circulatory support devices, TCBI was an independent predictor of all-cause and cardiovascular mortality [8]. In both populations, they revealed a similar predictive value compared to GNRI. However, the prognostic effect of TCBI in patients with ADHF has never been reported, and our present study is the first to assess the association between TCBI and long-term clinical outcomes in patients with ADHF. Focusing on other nutritional indicators, GNRI was also not associated with cardiovascular mortality in the present study and, to our knowledge, only one study using PNI as a nutritional index was reported to detect an independent association between the nutritional index and long-term cardiovascular mortality [30]. Our group previously showed the relationship between TCBI and cancer-related deaths in patients with coronary artery disease. In our previous study, although TCBI and GNRI correlated modestly, TCBI was shown as an independent predictor for cancer mortality [7]. Similarly, there was a modest correlation between TCBI and GNRI in the present study. However, GNRI was presented as an independent predictor for the cancer mortality in ADHF patients. This is consistent with the fact that malnutrition assessed by GNRI has been reported to be associated with cancer mortality in patients with malignancy [9,10].

In the present study, the cutoff value of TCBI to determine all-cause mortality in patients with ADHF was determined. However, such cutoff values may vary according to the patient population. Thus, the cutoff value may not be applicable in patients with other cardiovascular diseases and ADHF patients from other countries. Considering the results of analyses regarding NRI, TCBI provides additional discriminatory power for all-cause mortality, and GNRI provides additional discriminatory power for cancer mortality. These findings suggest that the assessment of TCBI can be clinically valuable, especially in the view of long-term ADHF care. The fact that TCBI can be obtained based on easier calculation than other nutritional indices may further support this.

This study has had several limitations. First, it was conducted as a single-center retrospective observational study and involved a limited number of patients. Second, we could not exclude the possibility that unmeasured factors could have influenced some of our findings, even after confounding factors were considered. Third, it was difficult to assess residual fluid retention after diuretic therapy, which is a unique hallmark of HF, and we could not exclude its influence on BW, TC, and TG. Furthermore, we may have to consider the effects of the short-term variability of TG. However, because laboratory data including TG were obtained in a fasting state within 24 h from hospital admission in the present study, such an effect of short-term variability of TG may be minimal. In addition, although using TCBI as a nutritional indicator may overcome the individual limitations of each indicator, the association between the changes in each indicator during hospitalization and clinical outcomes should be assessed in a further study.

## 5. Conclusions

The novel and simply calculated nutritional indices—TCBI and GNRI—may be differently interpreted for nutritional assessment in hospitalized patients with ADHF. However, their prognostic value may be similar in terms of long-term all-cause and cancer mortalities. Because TCBI is based on easier calculation than GNRI, TCBI can be used as an alternative to assess the nutritional status in the ADHF population.

## Figures and Tables

**Figure 1 nutrients-12-03311-f001:**
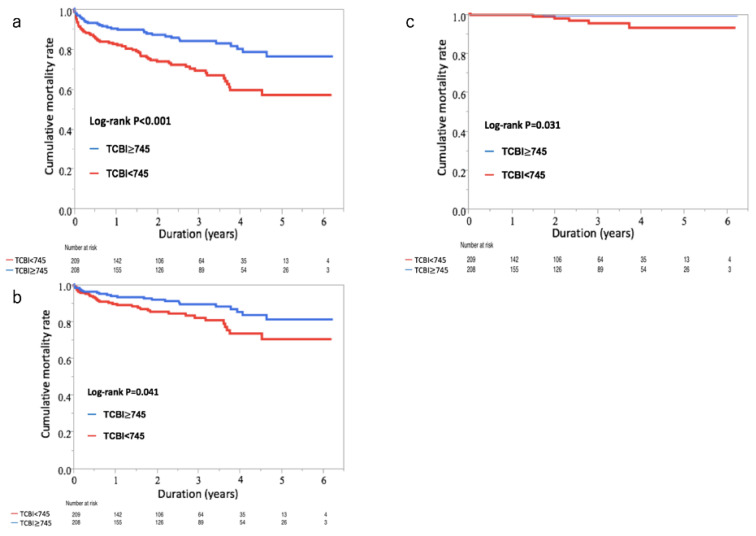
Cumulative event-free survival curves of all-cause, cardiovascular, and cancer deaths in patients with acute decompensated heart failure (ADHF). (**a**) Cumulative event-free survival curves of all-cause deaths in patients with ADHF. In the lower TCB index (TCBI) group, the cumulative incidence of all-cause deaths significantly increased compared with the higher TCBI (log-rank test: *p* < 0.001). (**b**) Cumulative event-free survival curves of cardiovascular deaths in patients with ADHF. In the lower TCBI group, the cumulative incidence of cardiovascular deaths significantly increased compared with the higher TCBI (log-rank test: *p* < 0.041). (**c**) Cumulative event-free survival curves of cancer deaths in patients with ADHF. In the lower TCBI group, the cumulative incidence of cancer deaths significantly increased compared with the higher TCBI (log-rank test: *p* < 0.031).

**Figure 2 nutrients-12-03311-f002:**
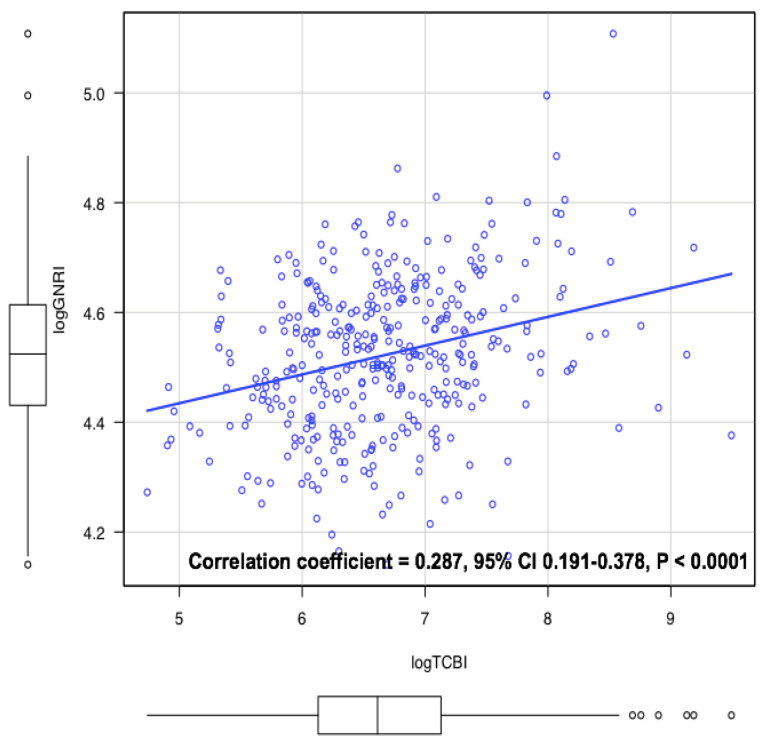
The correlation between TCBI and Geriatric Nutritional Risk Index (GNRI). TCBI was mildly correlated with GNRI (correlation coefficient = 0.287, 95% confidential interval (CI) 0.191–0.378, *p* < 0.0001).

**Figure 3 nutrients-12-03311-f003:**
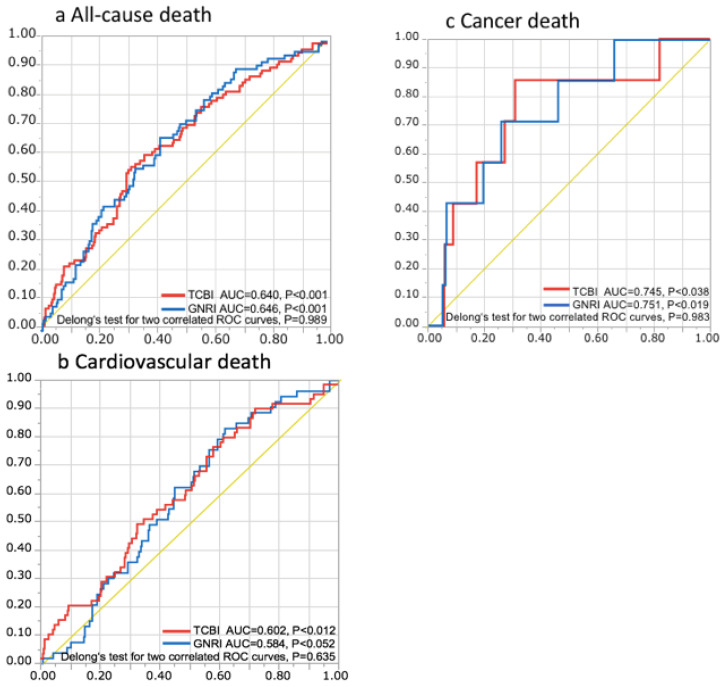
Prognostic implications of TCBI and GNRI for all-cause, cardiovascular, and cancer deaths. Receiver operating characteristic (ROC) curves of TCBI and GNRI with reference line for all-cause (**a**), cardiovascular (**b**), and cancer deaths (**c**). AUC: area under the curve.

**Table 1 nutrients-12-03311-t001:** Baseline characteristics of all patients.

	TCBI < 745 *n* = 209	TCBI ≥ 745 *n* = 208	*p*
Age, year	74.2 ± 11.0	65.7 ± 14.2	<0.001
Gender (male), *n* (%)	123 (58.8)	147 (70.6)	0.013
BMI, kg/m^2^	21.2 ± 3.7	24.6 ± 5.4	<0.001
BW, kg	52.7 ± 12.4	65.8 ± 18.5	<0.001
NYHA class II, (%)	30 (14.3)	43 (20.6)	0.228
III, (%)	76 (36.3)	68 (32.6)	
IV, (%)	103 (49.2)	97 (46.6)	
Ischemic heart disease, *n* (%)	95 (45.4)	91 (43.7)	0.767
History of heart failure, *n* (%)	108 (56.6)	105 (50.4)	0.844
AF, *n* (%)	94 (44.9)	91 (34.1)	0.027
Diabetes mellitus, *n* (%)	61 (29.1)	90 (43.2)	0.003
ICD, *n* (%)	4 (1.9)	4 (1.9)	1.000
CRT, *n* (%)	1 (0.4)	1 (0.4)	1.000
Mean BP, mmHg	95.3 ± 19.1	101.1 ± 23.6	0.011
Systolic BP, mmHg	136.3 ± 29.4	140.8 ± 33.8	0.182
Diastolic BP, mmHg	74.7 ± 17.3	81.4 ± 21.4	0.001
HR, rate/min	93.1 ± 29.4	96.7 ± 27.7	0.239
LVEF, %	44.3 ± 17.9	40.3 ± 16.0	0.018
Hemoglobin, g/dL	11.8 ± 2.2	13.2 ± 2.4	<0.001
eGFR, mL/min/1.73 m^2^	53.9 ± 27.1	53.9 ± 23.3	0.991
BUN, mg/dL	26.1 ± 14.7	26.2 ± 17.0	0.934
Sodium, mmol/L	138.2 ± 4.4	139.2 ± 3.8	0.010
Potassium, mmol/L	4.2 ± 0.6	4.2 ± 0.6	0.269
TC, mg/dl	147 ± 31.3	191 ± 41.7	<0.001
TG, mg/dl	64.2 ± 19.8	132.2 ± 71.6	<0.001
CRP, mg/dL	0.9 [3.3]	0.7 [2.9]	0.975
BNP, pg/mL	679 [793.4]	529.2 [698.1]	0.176
Medications at admission			
Beta blocker, *n* (%)	61 (29.1)	68 (32.6)	0.459
ACE-Is/ARBs, *n* (%)	74 (35.4)	83 (39.9)	0.364
Aldosterone blocker, *n* (%)	35 (16.7)	19 (9.1)	0.028
Diuretics, *n* (%)	87 (41.6)	62 (29.8)	0.014
Statin, *n* (%)	51 (24%)	38 (18.3%)	0.951
GNRI	89.5 [17.2]	94.3 [17.7]	<0.001

Variables are expressed as the mean ± standard deviation, median [interquartile range] or *n* (%). AF: atrial fibrillation, ACE-I: angiotensin-converting enzyme inhibitor, ARB: angiotensin II receptor blocker, BMI: body mass index, BW: body weight, BNP: B-type natriuretic peptide, BUN: blood urea nitrogen, BP: blood pressure, CRP: C-reactive protein, CRT: cardiac resynchronization therapy, eGFR: estimated glomerular filtration rate, GNRI: geriatric nutritional risk index, HF: heart failure, HR: heart rate, ICD: implantable cardioverter defibrillator, LVEF: left ventricular ejection fraction, NYHA: New York Heart Association, TC: total cholesterol, TG: triglyceride.

**Table 2 nutrients-12-03311-t002:** Results of univariable analysis and final model of multivariable analysis using Cox proportional hazard analysis of all-cause (a), cardiovascular (b), and cancer deaths (c).

		Univariable		Multivariable	
		HR (95% CI)	*p*	HR (95% CI)	*p*
a. All-cause deaths	Age (1 year increase)	1.05 (1.04–1.07)	<0.001	1.02 (1.00–1.04)	0.031
	Gender, male	1.67 (1.11–2.51)	0.014		
	Ischemic heart disease, yes	1.77 (1.18–2.68)	0.006		
	AF, yes	1.49 (0.97–2.33)	0.070		
	Mean BP (1 mmHg increase)	0.99 (0.98–1.00)	0.017		
	Hemoglobin (1 g/dL increase)	0.73 (0.67–0.80)	<0.001	0.86 (0.78–0.95)	0.004
	Na (1 mmol/L increase)	0.92 (0.88–0.96)	<0.001	0.91 (0.87–0.95)	<0.001
	eGFR (1 mL/min/1.73 m^2^ increase)	0.97 (0.97–0.99)	<0.001		
	BUN (1 mg/dL increase)	1.02 (1.01–1.03)	<0.001	1.02 (1.00–1.03)	0.003
	History of HF, yes	2.30 (1.51–3.61)	<0.001		
	Diuretics, yes	2.48 (1.65–3.74)	<0.001	2.3 (1.45–3.72)	<0.001
	Log CRP (1 increase)	1.19 (1.05–1.35)	0.007		
	Log BNP (1 increase)	1.39 (1.13–1.72)	0.002		
	Log GNRI (1 increase)	0.03 (0.01–0.15)	<0.001		
	Log TCBI (1 increase)	0.51 (0.38–0.68)	<0.001	0.64 (0.44–0.94)	0.024
b. Cardiovascular deaths	Age (1 year increase)	1.05 (1.02–1.07)	<0.001	1.03 (1.00–1.06)	0.019
	Hemoglobin (1 g/dL increase)	0.73 (0.65–0.81)	<0.001	0.82 (0.72–0.93)	0.002
	Sodium (1 mmol/L increase)	0.93 (0.88–0.99)	0.034	0.93 (0.87–0.99)	0.026
	BUN (1 mg/dL increase)	1.02 (1.01–1.03)	<0.001		
	Diuretics, yes	3.97 (2.33–6.98)	<0.001	3.39 (1.91–6.21)	<0.001
	Aldosterone blocker, yes	2.34 (1.26–4.13)	0.009		
	Beta blocker, yes	1.79 (1.05–3.00)	0.032		
	Log BNP (1 increase)	1.64 (1.24–2.19)	<0.001		
	Log GNRI (1 increase)	0.09 (0.01–0.66)	0.018		
	Log TCBI (1 increase)	0.55 (0.38–0.79)	0.001		
c. Cancer deaths	Hemoglobin (1 g/dL increase)	0.71 (0.50–0.96)	0.028		
	LVEF (1 increase)	1.06 (1.01–1.12)	0.023	1.05 (1.0–1.12)	0.016
	Log GNRI (1 increase)	0.01 (0.01–0.15)	0.008	0.01 (0.01–0.1)	0.006
	Log TCBI (1 increase)	0.25 (0.08–0.77)	0.014		

AF: atrial fibrillation, BNP: B-type natriuretic peptide, BUN: blood urea nitrogen, BP: blood pressure, CRP: C-reactive protein, eGFR: estimated glomerular filtration rate, GNRI: geriatric nutritional risk index, HF: heart failure, LVEF: left ventricular ejection fraction.

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
