# Peer review of "Prognostic Effect of a Novel Simply Calculated Nutritional Index in Acute Decompensated Heart Failure"

_nutrients, 2020, doi:10.3390/nu12113311_

Round 1

Reviewer 1 Report

The manuscript “Prognostic effect of a novel simply calculated nutritional index in acute decompensated heart failure” is an original paper dealing with clinical application of a novel  index based on blood lipids and body weight as a prognostic factor in patients hospitalized with acute decompensated heart failure.

the paper was prepared by the group who developed and tested the index TCBI in preceding publications (in coronary artery disease and in critical care care patients requiring mechanical circulatory support). I could not identify the use of the index by other study groups.

Regarding the methodology, my main question refers to patients selection. I understand this is a “real-world” cohort but  the definition of acute decompensated heart failure is a little bit unclear considering that close to 20% of patients were in New York Heart Association class 2, and the average ejection fraction was about 40%. CRT and ICD use was close to none. Therefore a clarified definition of ADHF used in this study would be welcome.

The analysis of nutritional index data is clear and proves that the novel index is equivalent to the older GNRI index, somewhat more complex in terms of prognostic capacity. A question arises regarding the practical use of this index as no calculation of prognostic indices such as positive or negative prognostic value for mortality related to specific threshold of TCBI is discussed.

The discussion is well conducted however it would be interesting to know whether all three components of the index contributed equally to the prognostic power.

Author Response

Dear editor: 

Thank you for your consideration. We have read the reviewers’ comment and corrected accordingly.

Responses to the Reviewer 1

C1: Regarding the methodology, my main question refers to patients’ selection. I understand this is a “real-world” cohort but the definition of acute decompensated heart failure is a little bit unclear considering that close to 20% of patients were in New York Heart Association class 2, and the average ejection fraction was about 40%. CRT and ICD use was close to none. Therefore, a clarified definition of ADHF used in this study would be welcome.

R1: We enrolled ADHF patients defined based on modified Framingham criteria, which only includes variables estimated at admission in Framingham criteria.In the larger Japanese ADHF registry, ATTEND registry in which patients were diagnosed as ADHF with the same diagnostic definition during the same recruitment period (i.e. 2007-2011) was used (Am Heart J. 2010;159:949-955), the frequency of patients with NYHA class II and less was 17.3% which is almost same as ours (Circ J. 2013; 77: 944-951). In addition, frequency of CRT/ICD use was also low in the ATTEND registry (2.3%/3.4%, Circ J. 2013; 77: 944-951). We now added brief description of modified Framingham criteria and related references (P2, L78-79).

C2: The analysis of nutritional index data is clear and proves that the novel index is equivalent to the older GNRI index, somewhat more complex in terms of prognostic capacity. A question arises regarding the practical use of this index as no calculation of prognostic indices such as positive or negative prognostic value for mortality related to specific threshold of TCBI is discussed.

R2: We performed ROC analysis regarding all-cause death in order to determine the cutoff of TCBI, and found that TCBI of 578 with positive/negative predictive values of 0.844 and 0.341, respectively. We added methods and results of these analyses and brief comments regarding these findings in the Discussion section (P10, L304-307).

C3: The discussion is well conducted however it would be interesting to know whether all three components of the index contributed equally to the prognostic power. 

R3: We analyzed BW, TC and TG in univariable analyses for all-cause, cardiovascular and cancer deaths, and found that BW alone and TC alone were significant factors for all cause, cardiovascular and cancer deaths. When variables which p value less than 0.1 were included in multivariable analysis, three components of the index seem not contribute equally (see supplemental table below). To find out importance of TCBI rather than BW alone or TC alone, we assessed reclassification improvements when we added BW on TC and found in the analysis for all-cause death, BW with TC showed significant net reclassification improvement compared with TC alone (NRI: 0.62, p<0.001). Furthermore, we added TG on TC+BW, and TG with TC+BW showed further net reclassification improvement compared with TC+BW (NRI: 0.17, p=0.048). Considering these results, we believe composite of BW, TC and TG (i.e. TCBI) should be focused.

Univariable

Multivariable

HR (95% CI)

p

HR (95% CI)

p

All-cause deaths

BW

0.98 (0.93–0.96)

< 0.001

0.97(0.95-1.00)

0.057

TC

0.99 (0.98–1.00)

0.003

0.99(0.98-0.99)

0.004

TG

0.99 (0.99–1.00)

0.207

Cardiovascular deaths

BW

0.96 (0.94–0.98)

<0.001

TC

0.99 (0.98–1.00)

0.044

0.99(0.98-0.99)

0.026

TG

0.99 (0.99–1.00)

0.127

Cancer deaths

BW

0.90 (0.82–0.97)

0.004

0.89(0.78-0.98)

0.014

TC

0.97 (0.94–1.00)

0.015

0.97(0.94-0.99)

0.021

TG

0.97(0.97–1.00)

0.478

Reviewer 2 Report

Dear Authors

The manuscript is generally well written and follows the idea previously presented and published by Juntendo University research team. However, to prove the novelty of TCBI in ADHF, more deep insight into the obtained data/results and stronger arguments for additional value of these index are needed.

Comments

Methods

  1. Please clarify if patients with diagnosed cancer (during initial hospitalization) were excluded from the analysis.
  2. Please explain why cancer mortality was included in follow-up data collection?
  3. Please explain the genesis of abbreviation for “TCB”
  4. Please describe in detail laboratory methods for TG, TC, albumin.

Results

  1. Please provide data on frequency of statin (and other lipid lowering drugs) use in both compared groups. If statistically different – include in hazard models.
  2. Were TC, TG and BM investigated as a possible risk factors alone?
  3. Was GNRI included in multivariate hazard analysis which also included TCBI? In Table 3 there seems to be a “shorten” presentation of results for GNRI that was analyses separately in the manner as TCBI in Table 2, yes? The model including both TCBI and GNRI could show which index is independent.
  4. The AUC for TCBI and GNRI are not high. Were other risk factors from multivariable analysis (Table 2), such as i.e. haemoglobin, Na, tested by ROC analysis and compared with TCBI?
  5. The comparison with CONUT and PCI, calculated for the study group, would be desirable.
  6. The % of beta-blockers and ACEI/ARB use is quite low as for HF patients? Please explain the potential cause.

Discussion

  1. I can agree that TCBI presented similar prognostic power as GNRI but one should be careful to TCBI is “is much easier to obtain” than GNRI, CONUT, PCI. Serum albumin, lymphocyte count (and BMI) are also easy to obtain and frequently measured in patients with ADHF. From clinical point of view these parameters are even more important in acute setting (hypoalbuminemia for edema diagnosis, lymphocytes for infection diagnosis). Even if in retrospective analysis % lack of this data was higher than for TC/TG/BM, it is not equivalent with its lower accessibility.
  2. Overall, the discussion do not convince to use TCBI as an alternative to other nutrition indexes and clinical indices (i.e. haemoglobin) in prognosis in ADHF patients.
  3. The additional value which I recognize for TCBI is a possibility to identify patients with higher risk of cancer death. It would be of clinical value, only if initially patients with cancer were excluded.
  4. In limitations it should be revealed that in the formula of TCBI one parameter (TC) includes in its calculation other one (TG). The short-term variability of TG should be also commented.

Others:

  1. Page 2 verse 66 – please correct “HF preserved ejection fraction” to “HF preserved with ejection fraction”
  2. The discussion should be divided to more sections/paragraphs.
  3. In Table 1 BM, TC, TG should be completed.

Author Response

Dear editor: 

Thank you for your consideration. We have read the reviewers’ comment and corrected accordingly.

Responses to the Reviewer 2 

C4: Please clarify if patients with diagnosed cancer (during initial hospitalization) were excluded from the analysis.

R4: We had excluded patients having cancer at the beginning and we mentioned about it in Page2, Line 82.

C5: Please explain why cancer mortality was included in follow-up data collection?

R5: Our group has been investigating long term clinical outcome in patients with cardiovascular disease. Especially, relationships between lipid/inflammatory/nutrition parameters or their drugs and long-term clinical outcomes including cancer related deaths were of interest for us. Indeed, we published several papers regarding relationship between them and cancer related deaths (Int J Cardiol. 2007; 114: 210-217. Int J Cardiol. 2018; 262: 92-98. Circ J. 2019; 83: 630-636). Thus, we included cancer mortality in the follow-up data collection.

 C6: Please explain the genesis of abbreviation for “TCB”

R6: The genesis of abbreviation for “TCB” is presented as TG x TCx BW (TCB) index. We have added the underline in Data collection section.This was indicated in the “Data collection” section (P3, Line 98).

C7: Please describe in detail laboratory methods for TG, TC, albumin.

R7: TG was measured by “Enzyme colorimetric method”. TC was measured by “Enzyme method”. Albumin was measured by “Improved BCP method”. We have added this fact in Data collection section (P3, Line 96-98).

C8: Please provide data on frequency of statin (and other lipid lowering drugs) use in both compared groups. If statistically different – include in hazard models.

R8: We have added the result of statin usage in Baseline characteristics as n=51 (24%) in TCBI<745 and N=38 (18.3%) in TCBI≧745 (p=0.951). In univariable analyses: statin use was not associated with all-cause (p=0.920), cardiovascular (p=0.445), cancer death(p=0.609). We just added use of statins in Table 1.

C9: Were TC, TG and BM investigated as a possible risk factors alone?

R9: Yes, TC alone and BW alone were identified as risk factors for cardiovascular and cancer death, and TC alone was also identified as a risk factor for all-cause mortality. Please see following supplemental table regarding univariable, and multivariable analyses in which variables with p value less than 0.1 in the univariable analysis were included. To find out importance of TCBI rather than BW alone or TC alone, we assessed reclassification improvement when we added BWon TC. We found in all-cause death, TC+BWshowed statistically significant net reclassification improvement compared to TCalone (NRI: 0.62, p<0.001).Furthermore, we further added TGon TC+BWand found in all-cause death, TC+BW+TGshowed statistically significant net reclassification improvement compared to TC+BW(NRI: 0.17, p=0.048). Considering these results, we believe composite of BW, TC and TG (i.e. TCBI) should be used.

Univariable

Multivariable

HR (95% CI)

p

HR (95% CI)

p

All-cause deaths

BW

0.98 (0.93–0.96)

< 0.001

0.97(0.95-1.00)

0.057

TC

0.99 (0.98–1.00)

0.003

0.99(0.98-0.99)

0.004

TG

0.99 (0.99–1.00)

0.207

Cardiovascular deaths

BW

0.96 (0.94–0.98)

<0.001

TC

0.99 (0.98–1.00)

0.044

0.99(0.98-0.99)

0.026

TG

0.99 (0.99–1.00)

0.127

Cancer deaths

BW

0.90 (0.82–0.97)

0.004

0.89(0.78-0.98)

0.014

TC

0.97 (0.94–1.00)

0.015

0.97(0.94-0.99)

0.021

TG

0.97(0.97–1.00)

0.478

C10: Was GNRI included in multivariate hazard analysis which also included TCBI? In Table 3 there seems to be a “shorten” presentation of results for GNRI that was analyses separately in the manner as TCBI in Table 2, yes? The model including both TCBI and GNRI could show which index is independent.

R10: No, we did not include both GNRI and TCBI together. Answer to the second question is “Yes”. We now revised Table 3 and related sections based on the multivariable analyses including both GNRI and TCBI.

C11: The AUC for TCBI and GNRI are not high. Were other risk factors from multivariable analysis (Table 2), such as i.e. haemoglobin, Na, tested by ROC analysis and compared with TCBI?

R11: We computed the area under the ROC curves of hemoglobin and sodium regarding each outcome. The AUC of hemoglobin for all-cause death was greater than that of the TCBI (0.743 vs 0.626, P < 0.001). The AUC of hemoglobin for cardiovascular death was also greater than that of the TCBI (0.720 vs 0.581, P = 0.004). There is no significant difference in AUC for cancer death between hemoglobin and TCBI (0.718 vs 0.748, p=0.579). In terms of sodium, there are no significant differences in AUC for all-cause, cardiovascular and cancer deaths between sodium and TCBI (all-cause death: 0.604 for sodium vs 0.624 for TCBI, p=0.650; cardiovascular death: 0.570 for sodium vs 0.572 for TCBI, P = 0.971; cancer death: 0.670 for sodium vs 0.821 for TCBI, P = 0.212). We assessed reclassification improvement when we added TCBI on hemoglobin, sodium and both. We found in all-cause death, log (TCBI)+hemoglobin showed significant net reclassification improvement (NRI: 0.28, p=0.016). Furthermore, we found in all-cause death, log (TCBI) + hemoglobin + sodium showed significant NRI (NRI: 0.42, p<0.001). In cancer death, we found that log (TCBI) + hemoglobin showed significant net reclassification improvement (NRI: 0.72, p=0.036). Furthermore, we found in cancer death, log (TCBI) + hemoglobin + sodium showed significant NRI (NRI: 0.88, p<0.001) Thus, we believe that TCBI provides additional information in the risk stratification regarding all cause death and cancer death.

In terms of cardiovascular death model, we did not find reclassification improvement when we added TCBI on hemoglobin, sodium and both (NRI 0.15, p=0.288 for hemoglobin + log (TCBI) and NRI 0.05, p=0.706 for hemoglobin + sodium + log (TCBI), respectively).

We assessed reclassification improvement when we added GNRI on hemoglobin, sodium and both in cancer death because independent correlate to the cancer death was not TCBI but GNRI. In cancer death, we found that log (GNRI) + hemoglobin showed significant net reclassification improvement (NRI: 0.69, p=0.046). Furthermore, we found in cancer death, log (GNRI) + hemoglobin + sodium showed significant NRI (NRI: 0.96, p=0.002).

We added these results in the revised manuscript (P8, L227-234) with description regarding NRI in the Statistics section (P3, L128-130) and brief comments in the Discussion section (P10, L307-311).

C12: The comparison with CONUT and PCI, calculated for the study group, would be desirable.

R12: The data of lymphocyte count is not available in our dataset. Thus, we could not calculate CONUT and PNI score, unfortunately.

C13: The % of beta-blockers and ACEI/ARB use is quite low as for HF patients? Please explain the potential cause.

R13: In the present study, a half of patients does not have a history of heart failure (as shown in Table 1) who might not use beta blockers and ACE-I/ARB, and data regarding use of those medications was collected at the time of hospital admission. Thus, use of beta blockers and ACE-I/ARB was low. In the larger Japanese ADHF registry, ATTEND registry in which patients were enrolled during the same recruitment period to ours (i.e. 2007-2011), use of beta blockers and ACE-I/ARB is quite similar to ours (Circ J. 2013; 77: 944-951).

C14: I can agree that TCBI presented similar prognostic power as GNRI but one should be careful to TCBI is “is much easier to obtain” than GNRI, CONUT, PCI. Serum albumin, lymphocyte count (and BMI) are also easy to obtain and frequently measured in patients with ADHF. From clinical point of view these parameters are even more important in acute setting (hypoalbuminemia for edema diagnosis, lymphocytes for infection diagnosis). Even if in retrospective analysis % lack of this data was higher than for TC/TG/BM, it is not equivalent with its lower accessibility.

R14: We agreed with reviewer’s comment. We revised these expressions.

C15: Overall, the discussion do not convince to use TCBI as an alternative to other nutrition indexes and clinical indices (i.e. haemoglobin) in prognosis in ADHF patients.

R15:We believe that TCBI can be calculated easily and considering results of our additional statistics regarding NRI, TCBI provides additional information on the hemoglobin and sodiumand independent from GNRI (at least all-cause death) in terms of the risk stratification.We now revised these expressions (P8, Line 227-234 in Statistical analysis section, P5, Line 234-244 in Results section and P10, Line 307-311 in Discussion section). 

C16: The additional value which I recognize for TCBI is a possibility to identify patients with higher risk of cancer death. It would be of clinical value, only if initially patients with cancer were excluded.

R16: Thank you for your comment. As mentioned above, we had excluded the cancer patients at the beginning and we mentioned about it in Page2, Line 82.

C17: In limitations it should be revealed that in the formula of TCBI one parameter (TC) includes in its calculation other one (TG). The short-term variability of TG should be also commented.

R17: Please note that TC is measured independently from TG. Nevertheless, we have added brief comment regarding short-term variability of TG in the Limitation section (P10, L317-320).

C18: Page 2 verse 66 – please correct “HF preserved ejection fraction” to “HF preserved with ejection fraction”

R18: We have corrected “HF preserved ejection fraction” toHF with preserved ejection fraction.

C19: The discussion should be divided to more sections/paragraphs.

R19: We have corrected as you suggested.

C20: In Table 1 BM, TC, TG should be completed.

R20: We have included BW, TC, TG and TCBI in Baseline characteristics.

Round 2

Reviewer 2 Report

Thank you for considering my comments.

The manuscript was significantly improved.